# Genome-Wide Association Study for Udder Conformation Traits in Chinese Holstein Cattle

**DOI:** 10.3390/ani12192542

**Published:** 2022-09-22

**Authors:** Mudasir Nazar, Ismail Mohamed Abdalla, Zhi Chen, Numan Ullah, Yan Liang, Shuangfeng Chu, Tianle Xu, Yongjiang Mao, Zhangping Yang, Xubin Lu

**Affiliations:** 1College of Animal Science and Technology, Yangzhou University, Yangzhou 225009, China; 2Joint International Research Laboratory of Agriculture and Agri-Product Safety, Yangzhou University, Yangzhou 225009, China

**Keywords:** udder structure traits, SNPs, GWAS, FarmCPU, gene network analysis, Holstein cattle

## Abstract

**Simple Summary:**

Udder conformation traits are economically important for dairy animals in the dairy industry. Milk production loss can be reduced via a better udder structure in dairy cattle. Udder traits are related to milk production and the somatic cell count, which is a sign of mammary infections (mastitis); as such, it is vital to understand the genetic architecture underlying udder traits in Holstein Friesian cattle for genetic development and long-term selection. Through a GWAS on udder structure traits in Chinese Holstein cattle, we identified several significant single-nucleotide polymorphisms (SNPs) and candidate genes associated with udder traits. The results could provide useful information regarding the genetics architecture of udder structure traits, thus improving the genetic information, health, longevity, and production of dairy cattle.

**Abstract:**

Udder conformation traits are one of the most economic traits in dairy cows, greatly affecting animal health, milk production, and producer profitability in the dairy industry. Genetic analysis of udder structure and scores have been developed in Holstein cattle. In our research, we conducted a genome-wide association study for five udder traits, including anterior udder attachment (AUA), central suspensory ligament (CSL), posterior udder attachment height (PUAH), posterior udder attachment width (PUAW), and udder depth (UD), in which the fixed and random model circulating probability unification (FarmCPU) model was applied for the association analysis. The heritability and the standard errors of these five udder traits ranged from 0.04 ± 0.00 to 0.49 ± 0.03. Phenotype data were measured from 1000 Holstein cows, and the GeneSeek Genomic Profiler (GGP) Bovine 100 K SNP chip was used to analyze genotypic data in Holstein cattle. For GWAS analysis, 984 individual cows and 84,407 single-nucleotide polymorphisms (SNPs) remained after quality control; a total of 18 SNPs were found at the GW significant threshold (*p* < 5.90 × 10^−7^). Many candidate genes were identified within 200kb upstream or downstream of the significant SNPs, which include *MGST1*, *MGST2*, *MTUS1*, *PRKN*, *STXBP6*, *GRID2*, *E2F8*, *CDH11*, *FOXP1*, *SLF1*, *TMEM117*, *SBF2*, *GC*, *ADGRB3*, and *GCLC*. Pathway analysis revealed that 58 Gene Ontology (GO) terms and 18 Kyoto Encyclopedia of Genes and Genomes (KEGG) pathways were enriched with adjusted *p* values, and these GO terms and the KEGG pathway analysis were associated with biological information, metabolism, hormonal growth, and development processes. These results could give valuable biological information for the genetic architecture of udder conformation traits in dairy Holstein cattle.

## 1. Introduction

Udder conformation traits play a vital economic role in dairy cattle [1]. The udder traits of the cattle is one of the most important factors that can be used to estimate production performance [2]. Udder composite values were included in official national genomic evaluation systems to account for the impact of cow conformation on health traits in 2009 [3]. The udder structure traits are related to production losses [4] as a result of mastitis [5] and early culling of cattle [6]. A previous study showed that the selection for better udder condition reduced bovine mastitis and the somatic cell count in Danish Holstein cattle [7]. The structure of the udder is related to long-term milk production mastitis resistance [8]. Udder structure traits affect the mammary gland health and have been detected as phenotype traits for improving clinical mastitis resistance in dairy cattle [9]. The teat canal length measurements and the number of lactations are important parameters for udder health in dairy cattle [10]. Udder conformation traits are heritable and could be useful as phenotypes due to the decrease in mastitis in Holstein cattle [11]. The heritability of udder traits was reported at 0.23 and 0.32 for udder depth and teat length, respectively [12]. The heritability of udder traits ranged between 0.09 and 0.25 in Brown Swiss cattle [13]. In Czech Holstein cattle, genetic correlations between the current linear traits ranged from 0.75 between fore udder attachment and udder depth to 0.70 between rear udder height and rear udder width [14]. Bovine mammary infection is a serious risk, due to the fact of udder health problems that affect financial losses, wellbeing, and the production of dairy cattle [15]. Udder health can be improved using udder conformation traits [16]. Udder health is vital in dairy farming, as it is the foundation for cost-effective and clean milk production [10].

The identification of quantitative trait loci (QTL) is an essential step in identifying and understanding genetic variants associated with economically significant phenotypes. GWAS has become a widely used method for identifying QTL and genome regions associated with phenotypes in Holstein cattle [17]. GWAS has proven to be an effective tool for identifying genetic variant associated with economically important traits such as milk production [18], milk composition [19], fatty acid [20,21], milk protein composition [22], body conformation [23], reproduction [24], pigmentation [25], fertility [26] and mastitis [27,28]. GWAS has been performed on udder structure and teat size traits in Charolais cattle [29], and it has also been conducted for udder structure score and teat shape in Nellore-Angus crossbred cows [30]. A previous study revealed 12 QTL that control the different features of the mammary gland’s morphology in the Fleckvieh cattle population [31]. Gene Ontology (GO) terms, KEGG pathway enrichment analysis, and gene networks refer to a better understanding of the biological mechanisms underlying the development traits and identify potential candidate genes [32,33]. Ten candidate genes were reported for udder structure traits, including *ESR1, FGF2, FGFR2, GLI2, IQGAP3, PGR, PRLR, RREB1, BTRC*, and *TGFBR2,* across three French dairy cattle breeds [33]. There are very few GWAS reports on mammary system performance in Chinese Holstein cows. Therefore, our research aimed to find SNPs and candidate genes associated with udder structure conformation traits, anterior udder attachment (AUA), central suspensory ligament (CSL), posterior udder attachment height (PUAH), posterior udder attachment width (PUAW), and udder depth (UD) in Chinese Holstein cattle. Our results enhance the knowledge of useful biological information regarding the genetic architecture of these udder traits., these identified SNPs and candidate genes would become helpful resources for the genetic selection in Holstein cattle.

## 2. Materials and Methods

### 2.1. Ethics Statement

Whole processes for hair sample collection and data collection of phenotype traits were measured according to the plan proposed by the Ministry of Agriculture of the People’s Republic of China and the China Council on Animal Care. The Institutional Animal Health Care, and this study was also accepted by the Yangzhou University Animal Researchers Ethics Committee (Permit serial: SYXK (Su) IACUC 2012-0029). During the research, no animals were harmed.

### 2.2. Phenotype and Genotype Data Collection

The studied experimental population consisted of 1000 Holstein cows raised from 4 dairy farms in Jiangsu Province, China (Sihong Aideweigang Dairy Farm 1: 199; Xuyi Weigang Dairy Farm 2: 214; Xuzhou Yonghao Dairy Farm 3: 224; Huaxia Animal Husbandry Farm 4: 363 cattle). The hair samples for genotyping data were collected within two months (November–December, 2019). The five udder structure traits were measured individually by scores ranging from 1 to 9 points. The five udder structure traits consisted of the anterior udder attachment (AUA), central suspensory ligament (CSL), posterior udder attachment height (PUAH), posterior udder attachment width (PUAW) and udder depth (UD) of 1000 cows, were measured according to China National Standard (GB/T 35568-2017). Three expert technicians completed the phenotypic measurement of udder straits for each cow, and hair samples from the tail of 999 cattle were also collected for DNA extraction and genotyping.

### 2.3. Phenotypic and Genotypic Parameters

Statistical analysis and Pearson correlation of the phenotype traits were investigated using the computer software, SPSS (v.19.0, Chicago, IL, USA), for data analysis (mean, standard error, minimum and maximum scores, standard deviation, coefficient variance %, and phenotypic correlation of traits). Genetic analysis was carried out with an animal model using the DMU software (v.5.6) [34] to determine the heritability, and the genetic correlation between pairs of traits, as shown in the following equation:(1)$$y_{ijklm}=u+Herd_{i}+Year_{j}+Season_{k}+Parity_{i}+a_{m}+e_{ijklm}$$
where ***y_ijklm_*** is the phenotypes in the year ***j***th; ***k***th is the season; ***I***th is the individual parity ***m***th from the herd of animals ***i***th; ***u*** is the mean of the population; animal ***herd_i_*** is the herd effect according to a cows origin from one of the four herds; ***Year_j_*** is the year levels of the effects in ***j***th; ***Season_k_*** is the season levels of the effects ***k***th and parity is the effect of parity ***l***th; ***a*** is the additive effect of the individual ***m***th, which was conducted by the pedigree information; and ***e*** is the residual in the ***j***th year, ***k***th season, and ***l***th parity of the individual ***m***th from the ***i***th herd. All the effects were evaluated as fixed effects except for the overall mean. The pedigree data of the cows age could be traced back at least 3 generations (2009–2020), the parities of the cattle were from 1 to 4, and the 4 seasons represented “September–November, December–February, March–May, and June–August”.

### 2.4. Genotypiing Data and Quality Control

Genome DNA was extracted and genotyped using the GGP Bovine 100k SNP Chip, by Neogen Corporation (http://www.neogenchina.com.cn/; accessed on 25 June 2022) based on ARC-UCD1.2/bosTau9 as the genome reference. The GGP Bovine 100K SNP chip consisting of 100,000 SNPs was used for genotyping individuals. Then, quality control was conducted using Plink software (v.1.90, MA, USA, Cambridge) [35], to remove the markers that did not comply with the following standard: (1) the individual call rate lower than 95%; (2) the genotyping call rate of single SNP lower than 90%; (3) the minor allele frequency (MAF) of SNP > 0.05; (4) deviated from the Hardy–Weinberg equilibrium (*p* < 1.0 × 10^−6^). After quality control, there remained 984 cattle and 84,407 markers (SNPs) variants for further analyses. The distribution of the SNPs information on 29 chromosomes within 1mb window size shown in Appendix A.

### 2.5. Principal Component Analysis

For the analysis of the population structure, Plink 1.90 software (v1.90) [35] was used to perform PCA (principal component analysis) on 984 cattle genotypes with 84406 SNPs and whole information genome. The ggplot2 package in R (v.4.0.4) was used to visually analyze the PCA plot.

### 2.6. Association Analysis

This study used the multi-locus linear mixed model to perform the GWAS analysis between the SNPs and trails using the fixed and random model circulating probability unification (FarmCPU) method [36]. The FarmCPU method performed marker tests with associated markers as covariates in a fixed-effects model, and in a random-effects model, iteratively [36]. In order to obtain a convenient illustration, the names of the associated markers detected in the fixed-effects model at each cycle were called pseudo-quantitative trait nucleotides (QTNs). Pseudo-QTNs were used to define the kinship of individuals to avoid a model over-fitting problem in the fixed-effect model [36]. Population stratification is a critical component that can lead to false-positive results in association studies [37]. Consequently, the present study fitted the highest two principal components Principal (PCs) as covariate variables in the GWAS models. The fixed-effects model was as follows [36].
(2)$$y_{i\;}={\;M}_{i1}b_{1}+{\;M}_{i2}b_{2}+\cdots +M_{it}b_{t\;}+{\;S}_{ij}d_{i}+{\;e}_{i}$$
where y*_i_* is the *i*th of an individual: ***M*_i_**_1_, ***M_i_***_2_,……, ***M_i_****_t_* are the genotypic of the pseudo-QTNs, started with an empty set; ***b*_1_**, ***b*_2_**,……, ***b_t_*** are the consistent effects of the pseudo-QTNs; ***S_ij_*** is the genotype of the *i*th individual and *j*th genetic marker (SNPs); d*_j_* is the corresponding effect of the ***j***th genetic marker; ***e_i_*** are the residuals having a distribution with a zero mean and a variance of ***ϭ*^2^*_e_***.

Each marker (SNP) had its own *p* value after substitution. The *p* values and the associated marker map were used to update the selection of the pseudo-QTNs using the SUPER algorithm [38] in a REM as follows:$$y_{i}={\;u}_{i}+{\;e}_{i}$$
where ***y_i_*** and ***e_i_*** are similar to in Equation (1), and ***u_i_*** is the individual’s ***i***th overall genetic effect. The type I error false-positive rate was measured at the level of 5%, and the genome-wide significance of the threshold value was revealed according to this method (0.05/SNPs) after quality control, where Nsnps were the remaining number of SNPs after quality control [39]. After Bonferroni correlation, the significant threshold value for the genome-wide association study was *p* < 5.90 × 10^−7^ (0.05/84407). Then, the Manhattan plots and quantile–quantile plots (QQ plots) were drawn using the “CMplot” package in the R 4.1.0 software [40].

### 2.7. Annotation of Candidate Genes

All of the genes within a 200 kilobases (Kb) position of the significant SNPs were identified as candidate genes for udder traits. Two hundred kilobases is a common distance that has been used to annotate and find genes related to SNPs in previous association [41,42]. We identified genomic region and candidate gene through UCSC Genome Browser with the help of cow assembly in April 2018 (http://genome.ucsc.edu/bosTau9, ARS-CUCD1.2) accessed on 25 June 2022 [43].

### 2.8. Functional Pathway-Enrichment and Gene Network Analysis of Candidate Gene

In the study, to further understand the biological information among these candidate genes, we submitted all the candidate genes identified from the GWAS analysis into the DAVID database (Database for Visualization, Annotation and Integrated Discovery) software [44] for Gene Ontology terms (GO) [45] and KEGG pathway analysis [46]. *p* values were adjusted for multiple tests using the Bonferroni correction. Gene network analysis was carried out among genes using online software for the identification of Interrelating Genes (STRING database software v.11.0) [47] and Cytoscape software v.3.8.1 (cytoscape.org) [48] to visualize the analysis of the results.

## 3. Results

### 3.1. Descriptive Statistical Data Analysis

The adjusted udder conformation traits of 984 cattle included the anterior udder attachment (AUA), central suspensory ligament (CSL), posterior udder attachment height (PUAH), posterior udder attachment width (PUAW), and udder depth (UD) are presented for normal distribution in this study. This descriptive statistical data analysis of the traits is shown in Table 1. The frequency distribution of the adjusted phenotype of udder traits is shown in Figure 1.

### 3.2. Phenotypic and Genetic Correlations, and Heritability Estimation of Udder Traits

The phenotypic and genetic correlations between these traits are shown in Table 2. In the phenotypic correlation, the CSL trait was found to be positively correlated with PUAW and UD, and the other traits were negatively correlated with CSL. The CSL trait was genetically positively associated with PUAW, and other traits were genetically negative correlated. In contrast, the AUA trait was phenotypically positively correlated with CSL, PUAW and UD except PUAH. The distribution and phenotypic correlation among the udder traits are shown in Appendix A. The AUA trait was genetically positively correlated with PUAH and UD, and the other traits were negatively correlated. The PUAH trait was genetically positively correlated with PUAW and AUA, and it was phenotypically negative correlated. The PUAW trait was genetically highly negatively correlated with AUA and UD, and other traits were found to have low to moderate genetically positive correlation, while the PUAW trait was found to be positively phenotypic correlated with all traits. The genotypic correlation among the udder traits is shown (Appendix A). Moreover, the heritability and the standard errors were estimated for the udder traits at 0.24 ± 0.02, 0.34 ± 0.03, 0.04 ± 0.00, 0.13 ± 0.03, and 0.49 ± 0.03, respectively (Table 2).

### 3.3. Population Structure

This research used Principal Component Analysis to visualize the family structure. The result showed that all the populations were divided into two sized clusters group. The PCA results in Figure 2 identified that the first two principal components PCA1 and PCA2 participated in 11.8% and 9.2%, respectively; at the same time, the total variation was approximately 21%; therefore, the FarmCPU model was used to fit the first two PCs in the PCA association analysis as covariate variables. The mixed linear model considered several population stratifications based on the results of the PCA analysis. Two groups were represented in the population structure (Figure 2).

### 3.4. Genome-Wide Association Study

The FarmCPU model was used in the current study to conduct the genome-wide association analysis. The quantile–quantile (QQ) plots demonstrated that the genome-wide association study analysis model was reasonable in this research (Figure 3). The lambda values (λ) for the AUA, CSL, PUAH, PUAW and UD were 0.91, 1, 1.03, 0.94, and 1.05, respectively, and they were all close to 1.06. The red dot at the top right corner of the QQ plots also displayed the significant SNPs that were associated with the udder conformation traits in this study (Figure 3), and the population stratification was sufficiently controlled. The Manhattan plots reveal the results of the GWAS significance levels (−log10 of the *p*-value of each SNP) by chromosomes position; each chromosome showed a different color. Significant SNPs in the Manhattan plot were strongly associated with traits (Figure 4a). A Circular Manhattan plot for significance log−10 (*p* values) of the associated SNPs were shown relations to five udder traits. The five circles from outside to inside represent the udder conformation traits. (Figure 4b).

As stated before, the threshold value was 5.90 × 10^−7^ for significant SNPs in the GWAS. The 18 SNPs passed the threshold and were significantly associated with five udder traits (AUA, CSL, PUAH, PUAW, and UD), four SNPs (DB-340-seq-rs208014256, Hapmap58214-rs29015775, BovineHD2700005329, and BovineHD0900028603) positioned on chromosomes 5, 22, 27 and 9 were identified to be associated with the trait AUA. Three SNPs (ARS-BFGL-BAC-29174, Hapmap32447-BTC-033214, and BovineHD0600005127, located on chromosomes 21, 6, and 6, respectively) were identified to be associated with the trait CSL. For PUAH traits, three SNPs were presented on chromosomes 29, 18, and 22 (BovineHD2900000083, BovineHD1800011193, and BovineHD2200002408, respectively) Moreover, three significantly associated SNPs (BovineHD0700028083, BovineHD0500010522, and BovineHD1500023322) located on chromosomes 7, 5, and 15, respectively, were indicated to be associated with the trait PUAW. While five SNPs (BTA-75047-no-rs, BovineHD0600024277, BovineHD0600001885, BovineHD0900001933, and BovineHD2300001734) detected on chromosomes 5, 6, 6, 9, and 23, respectively, were found to be associated with the trait UD (Table 3).

### 3.5. Annotation of the Candidate Genes

In our study, genes that were identified at a distance within 200 kb of significant SNPs were recognized as candidate genes. Among all of the 18 significant SNPs associated with udder conformation traits, 14 of them were located within the following genes: Microsomal Glutathione S-Transferase 1 (*MGST1*); LOC101903734, Microtubule-Associated Scaffold Protein 1 (MTUS1), *PRKN* (Parkin RBR E3ubiquitin protein ligase); *GRID2* (Glutamate Ionotropic Receptor Delta-Type Subunit 2); LOC112447118, Forkhead Box P1 (*FOXP1*), SMC5-SMC6 Complex Localization Factor 1 (SLF1); Transmembrane Protein 117 (*TMEM117*), SET Binding Factor 2 (*SBF2)*; Galectin 2 (*LGALS2*); *GC* Vitamin D Binding Protein (*GC*); *ADGRB3* (Adhesion G Protein-Coupled Receptor B3); *GCLC* (Glutamate-Cysteine Ligase Catalytic Subunit). Although, one of the SNPs on Chr6 was located near (50 kb) to the Ubiquitin-Conjugating Enzyme E2 K gene (*UBE2K*), three SNPs on chromosomes21, Chr29, Chr18 were located closely (100 kb) to the *STXBP6* (Syntaxin-Binding Protein 6), *E2F8* (E2F Transcription Factor 8), and Cadherin 11 (*CDH11*), respectively (Table 3).

### 3.6. Enrichment Analysis

The Asian server used the NCBI database and UCSC Genome Browser by using cattle assembly in April 2018 (ARC-UCD1.2/bosTau9), and a total of 141 candidate genes were identified within the 200 kb region upstream/downstream of the significant SNPs for the udder structure traits; all of the candidate genes were used for KEGG pathway and enrichment analysis (GO). The results of the GO analysis determined 58 Gene Ontology terms (Appendix A and Figure 5).

The KEGG pathways analysis revealed 18 pathways (i.e., bta04978: mineral absorption involved three genes; bta00480: glutathione metabolism comprises three genes; bta00340: histidine metabolism had two genes; bta04935: growth hormone synthesis, secretion and action involved three genes) which are shown in Table 4, according to the adjusted *p* value. A KEGG pathway analysis of the dot plot is presented in the Appendix A.

### 3.7. Gene Network Analyses

The STRING database was used to perform a protein–protein interaction network using all the genes previously used in functional analysis. From the relationships between genes identified in this gene network, there were many interactions among the genes (consisting of 44 nodes related via 75 edges). The corresponding interactivity power among these genes was determined by the staining strength between the lines that linked one gene to the other (Figure 6).

## 4. Discussion

In this research, the heritability of the udder traits was low and medium between 0.04 and 0.49 as shown in Table 2. The results of the heritability of udder traits were reliable with the predictions by Wu et al. [49], who described the estimation of heritability of udder structure traits ranged from 0.08 to 0.22 in Chinese Holstein Cattle, while previous studies have reported the heritability of udder structure traits 0.18 to 0.37 in dairy cattle [5,50,51].

The phenotypic and genotypic correlations result of the udder conformation traits were negatively low to medium at −0.37 to 0.26 and −0.62 to 0.52, respectively, for Chinese Holstein cows (Table 2); the results of the udder structure traits are in agreement with the findings of Němcová et al. [14], who stated that the conforming phenotypic correlations were medium at 0.23 to 0.46, and genotypic correlations among the udder traits range from 0.70 to 0.75 in Czech Holstein cattle. Udder balance (UB) and fore udder attachment (FUA) showed genetic correlations of 0.40, 0.63, and 0.39 in three different French breeds (Montbeliarde, Normande, and Holstein), respectively [33].

Population stratification was a significant causative factor in GWAS. When a GWAS analysis included samples with different genetic structures, the PCA procedure [52] could identify the population structure by sorting individuals into grouped ancestry based on their genetic structure. All of the research populations were divided into two groups: one group contained a large quantity of clusters, and the other consisted of a small number of clusters (Figure 2). This identifies that every group is clustered nearly together and has a genotypic relationship. The use of Holstein semen from different overseas countries may have resulted in the division of two groups. or the cows in these farms may contain blood from other breeds, as we found that Chinese Holstein cows can be registered which are a minimum of 87.5% Holstein blood (GB/T 3157 2008, Chinese Holstein Cattle). As we performed the population structure, the PCs were adjusted as covariates to the association study for proper population stratification. A previous study found that the inflation factor (λ) for colostrum and serum albumin concentrations was 0.983 and 1.004 in Chinese Holstein cattle [53], respectively, indicating that the appropriate model successfully corrected the population stratification [54]. After correcting the population structure, the lambda (λ) value should be close to 1 [55]. The deviation of the observed value from the expected value is near 1.06 on the QQ plot (Figure 3), and the lambda value (λ) showed 0.94 < 1.05, both representing that the population stratification was adequately adjusted with a suitable method.

Population stratification is a major challenge during GWAS work analysis. Furthermore, population stratification can lead to many false positives in GWAS results due to incorrect associations [56]. Due to systematic ancestry variations in GWAS, population stratification is a significant confounding element that can lead to false positives [57]. Because there are numerous approaches for correcting population stratification, a statistical model can be a helpful tool for correcting and reducing the chances of type 1 error (false positive association) [58]. The FarmCPU method was used in our study for its benefits in fully controlling false positives, eliminating confusion, and improving the efficiency of computation by iteratively using a fixed-effect model, and random-effect models [36]. The QQ plots of the five traits are shown in Figure 4, and in this study, the effect of the cryptic interaction between the animals was effectively controlled.

In this research, GWAS revealed 18 significant SNPs (single-nucleotide polymorphism) associated with five udder structure traits in Holstein cattle using the FarmCPU model. Among all of the SNPs, two significant SNPs were ARS-BFGL-BAC-29174, located near the *STXBP6* gene, and BovineHD0700028083, located within *SLF1*. The *STXBP6* gene provides a syntoxin-binding protein which attaches to the SNARE complex components, preventing membrane fusions such as phagocytosis [59], and the *STXBP6* gene causes rheumatoid arthritis [60]. It has also been expressed in the breast cancer cells of mice [61]. The *SLF1* gene is also important in genome stability maintenance in human cells after DNA damage [62].

Our research identified genes associated with the AUA trait (i.e., *MGST1, MTUS1*, and *PRKN*). Some studies have determined that *MGST1* has the largest significant effect on Chr5 for fat production in U.S. Holstein cattle [17]. *MGST1* has been recognized extensively as a functional candidate gene for the QTL associated with milk composition features in Montbéliarde cows [63]. In two Holstein cattle breeds, the *MTUS1* gene was associated with meat quality and carcass traits [64]. The *PRKN* gene was associated with growth and production traits in Fuzhong Buffalo [65], and it was also associated with fatty acids in Chinese Wagyu cattle [66].

The Hapmap32447-BTC-033214 SNP located on Chr6; glutamate ionotropic receptor delta type subunit two genes (*GRID2*) is associated with the CSL trait, but, in the previous study, the *GRID2* was involved in the sexual maturity of Simmental cattle [67].

Furthermore, our results report that the *E2F8*, *CDH11*, and *FOXP1* genes are associated with the PUAH trait. E2F transcription factors, such as the *E2F8* gene control the expression of genes involved in cell cycle proliferation and progression [68]. The present study discovered a novel hormone, and developmental regulation of *E2F8* mRNA throughout bovine follicular growth [69]. The *CDH11* gene is involved in mastitis development and mammary gland growth in lactating cattle [70,71]. The *FOXP1* gene is important for mammary gland development during puberty and maturity in mice.

However, the results show that the *TMEM117*, and *SBF2* genes are associated with the PUAW trait. Zhu et al. [72] also conducted a GWAS and found that the *TMEM117* gene was related to saturated fatty acids composition in Simmental cattle. The *SBF2* gene has been identified by GWAS for the total solid yield trait in Thai dairy cows [73].

The BTA-75047-no-rs, BovineHD0600024277, BovineHD0900001933, and BovineHD2300001734 SNPs are located within genes (i.e., *LGALS2, GC, ADGRB3,* and *GCLC*, respectively), are associated with udder depth trait; meanwhile, one BovineHD0600001885 SNP located on Chr6 (50kb) was also related with this trait. Zexi et al. [74] reported that the GC gene has a pleiotropic effect that involves milk production and evidence of mastitis in Nordic Holstein cows. The result of the *GC* gene on milk production in Holstein cattle has already been investigated in association analysis [75]. The *ADGRB3* gene is responsible for the uniformity of yearling weight in Nellore cattle [76]. Functional pathway analyses of list candidate genes are shown; The *GCLC* gene is consistently increased as parturition approaches and lactation began in Holstein Dairy cattle [77], while our finding provides evidence that *GCLC* is associated with udder depth trait.

In our research, GO and KEGG pathways analysis identified many GO terms and KEGG pathways. These included significant candidate genes that are associated with udder traits under analysis, for example, Proteolysis, a GO:0006508 biological process which has nine genes in Appendix A, among which is *PDCD5*. The protein-containing complex subunit organization, GO:0043933, had eight genes including *UGDH, STXBP6*, and *FKBP1A*. The cellular amide metabolic process (GO:0043603) had seven genes involving *MGST2 and HAL.* Embryo development ending in birth (GO:0009792) consisted of four genes, including *E2F8*. Cardiac muscle tissue development (GO:0048738) consisted of two genes (*CSRP3,* and *FKBP1A*). All these identified genes were closely related to our significant SNPs. As mentioned above, some candidate genes that participated in the GO term may play an essential role in the biological function of body traits, and afterward, these candidate genes participate in udder traits. The *PDCD5* gene regulates cell proliferation, cell cycle progression, and apoptosis in bovine cancer cells (*A431*) [78]. The *MGST2* has been found as a potential gene in the bovine mammary gland [79]. The *UGDH* gene is associated with milk production traits in Chinese Holstein [80]. These four KEGG pathway analyses were enriched for udder traits; mineral absorption, glutathione metabolism, histidine metabolism, growth hormone synthesis, secretion and action. The minerals absorption pathway was significantly related to udder traits of dairy cow [81]. Glutathione metabolism affected by liver function due to high blood glutathione in early lactation of Holstein cows [82]. Histidine metabolism is involved in the mammary cell growth, protein synthesis, and milk yield of Holstein cows [83]. Prolactin is a growth hormone related to lactation and mammary growth in dairy cows [84]. The functional analysis resulted from several terms and pathways related to protein-containing compounds (amino acids), absorption, catabolism and metabolism, and liver growth. Hence, it was assumed that all significant SNPs and candidate genes might possibly be associated with udder conformation traits.

## 5. Conclusions

In conclusion, our research found 18 significant SNPs associated with udder conformation traits in Chinese Holstein cattle. Several candidate genes harbored SNPs (i.e., *MGST1*, *MGST2*, *MTUS1*, *PRKN*, *STXBP6*, *GRID2*, *E2F8*, *CDH11*, *FOXP1*, *SLF1*, *TMEM117*, *SBF2*, *GC*, *ADGRB3* and *GCLC*) identified mostly to participate in biological information, metabolism, and development processes. Our findings provide useful biological information for understanding the genetic architecture for improving udder traits and will therefore contribute to the genetic selection of Chinese Holstein cattle.

## Figures and Tables

**Figure 1 animals-12-02542-f001:**
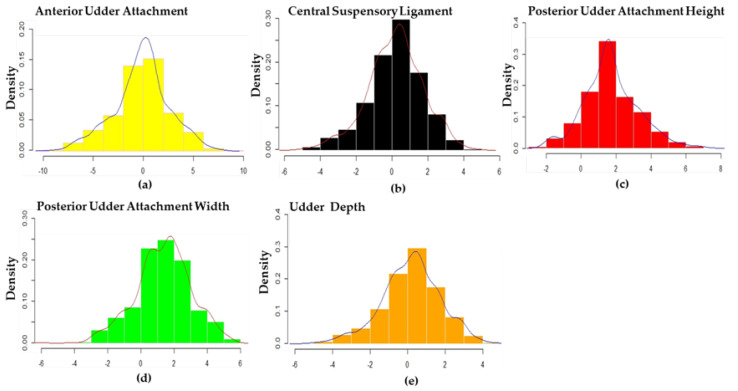
Distribution of the adjusted phenotype of AUA (**a**), CSL (**b**), PUAH (**c**), PUAW (**d**), and UD (**e**) in the population of Holstein cows. The adjusted phenotypes of the five traits shown about normal distribution.

**Figure 2 animals-12-02542-f002:**
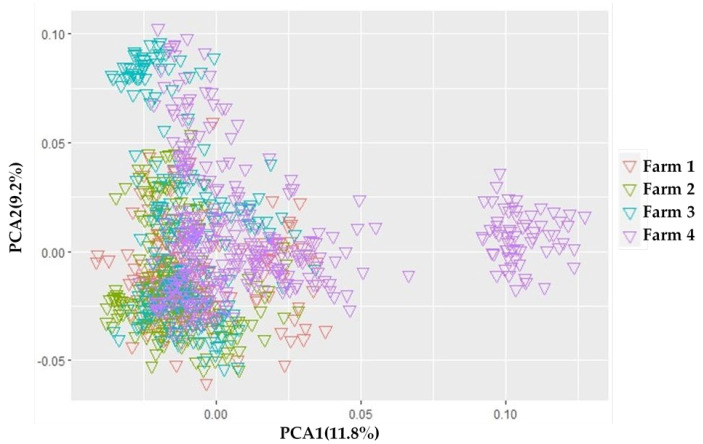
Population structure demonstrated by the 984 cattle raised at four animal farms. Principal component analysis (PCA).

**Figure 3 animals-12-02542-f003:**
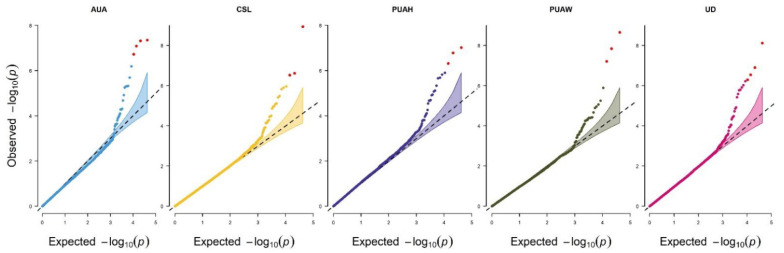
The quantile–quantile plots of the five udder traits from the GWAS in Holstein cow. The separation between the observed and expected values was analyzed using QQ plots. The null hypothesis indicated no relationship with the red lines. Deviation of the expected *p*-value showed that the population stratification was adequately controlled in the tails for every trait. The red dot significant SNPs indicate the threshold value.

**Figure 4 animals-12-02542-f004:**
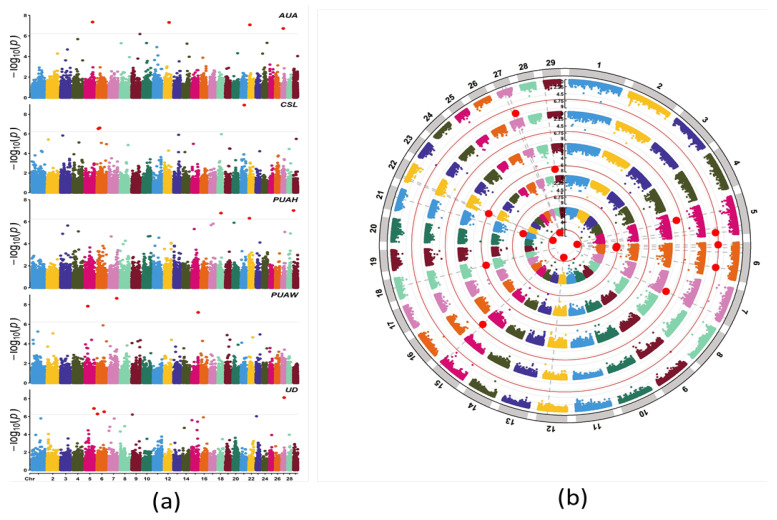
(**a**) Manhattan plots established from the GWAS results of the udder conformation traits in Holstein cattle. The significant threshold was *p* < 5.9 × 10^−7^. The Manhattan plot displays genomic SNPs on the horizontal axis (x-axis) along with chromosomes and the negative logarithm of each SNP’ association *p* value on the vertical axis (y-axis). After the Bonferroni correction, the green line shows a significant threshold level. (**b**) In the Circular Manhattan plot, the udder conformation traits were plotted from outside to inside, respectively.

**Figure 5 animals-12-02542-f005:**
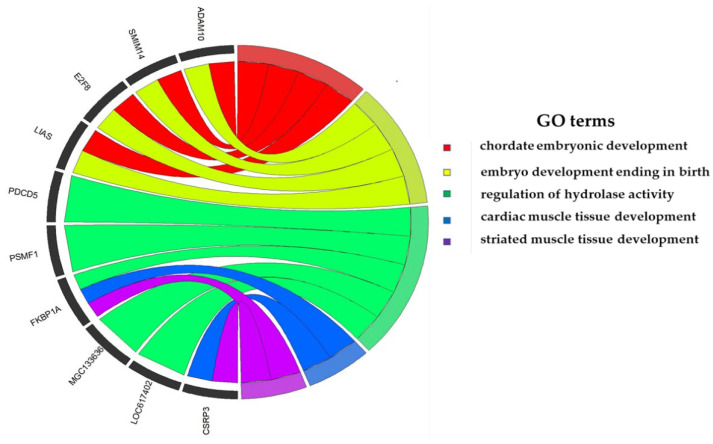
Gene Ontology term results from the udder conformation traits.

**Figure 6 animals-12-02542-f006:**
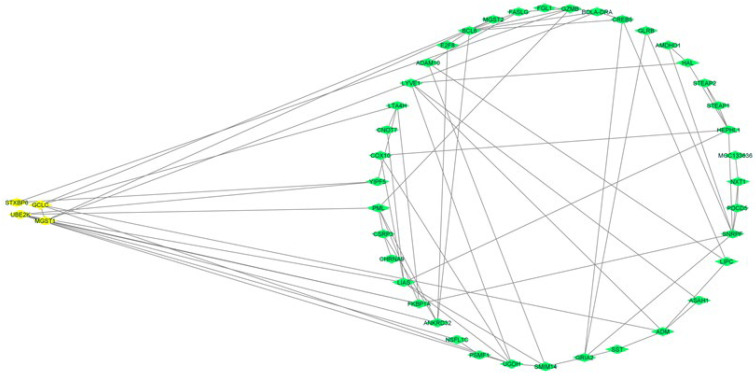
Gene network analysis for genes associated with udder structure traits in Holstein cow. (The yellow nodes represent the significant candidate genes and their interaction with related genes).

**Table 1 animals-12-02542-t001:** Descriptive statistics for the adjustment of udder traits of Holstein cows (984).

Traits	Mean	SE	Min	Max	SD	CV%	Skewness	Kurtosis	*p* Value
AUA	0.17	0.09	−10.15	8.04	2.81	7.94	0.22	0.32	2.53 × 10^−6^
CSL	0.45	0.04	−1.88	8.85	2.67	3.50	0.33	0.50	2.20 × 10^−7^
PUAH	1.73	0.05	−2.2	7.32	1.55	2.41	0.34	0.53	5.70 × 10^−8^
PUAW	1.34	0.05	−5.9	5.59	1.68	2.83	0.26	0.23	1.00 × 10^−4^
UD	0.18	0.05	−5.29	4.75	1.56	2.43	0.24	0.34	3.206 × 10^−5^

Mean; SE, standard error; Min, minimum; Max, maximum; SD, standard deviation; CV, coefficient of variation. AUA, anterior udder attachment; CSL, central suspensory ligament; PUAH, posterior udder attachment height; PUAW, posterior udder attachment width; UD, udder depth.

**Table 2 animals-12-02542-t002:** Genotyping (upper diagonal) and phenotyping (lower diagonal) correlations, and the heritability (grey, diagonal) for udder traits in Holstein cattle.

Traits	AUA	CSL	PUAH	PUAW	UD
AUA	0.24 (0.02)	−0.38	0.22	−0.62	0.10
CSL	0.02	0.34 (0.03)	−0.44	0.52	−0.45
PUAH	−0.11 **	−0.37 **	0.04 (0.00)	0.14	0.09
PUAW	0.17 **	0.14 **	0.002	0.13 (0.01)	−0.49
UD	0.26 **	0.14 **	−0.07 *	0.07 *	0.49 (0.03)

AUA, anterior udder attachment; CSL, central suspensory ligament; PUAH, posterior udder attachment height; PUAW, posterior udder attachment width; UD, udder depth. The upper subscript * and ** represent significant correlation at 0.05 and 0.01, respectively.

**Table 3 animals-12-02542-t003:** GWAS significant SNPs associated with udder structure conformation traits in Holstein cattle.

Traits	SNPs	CHR	Position (kb)	MAF	Nearest Gene	Distance (kb)	*p*-Value	Effect
AUA	DB-340-seq-rs208014256	5	93520616	0.46	*MGST1*	Within	4.48 × 10^−8^	0.330783
	Hapmap58214-rs29015775	22	13159539	0.49	*LOC101903734*	Within	8.34 × 10^−8^	−0.35043
	BovineHD2700005329	27	19594311	0.16	*MTUS1*	Within	1.90 × 10^−7^	−0.47118
	BovineHD0900028603	9	97665052	0.25	*PRKN*	Within	6.48 × 10^−7^	0.371653
CSL	ARS-BFGL-BAC-29174	21	40773446	0.43	*STXBP6*	100 kb	1.16 × 10^−9^	0.36344
	Hapmap32447-BTC-033214	6	32254947	0.42	*GRID2*	Within	2.45 × 10^−7^	−0.31119
	BovineHD0600005127	6	17417238	0.39	*LOC112447148*	Within	3.02 × 10^−7^	−0.31736
PUAH	BovineHD2900000083	29	702083	0.44	*E2F8*	100 kb	9.70 × 10^−8^	0.298361
	BovineHD1800011193	18	37485453	0.47	*CDH11*	100 kb	1.66 × 10^−7^	0.26997
	BovineHD2200002408	22	8008314	0.12	*FOXP1*	Within	4.89 × 10^−7^	−0.42465
PUAW	BovineHD0700028083	7	93970405	0.38	*SLF1*	Within	2.26 × 10^−9^	−0.33939
	BovineHD0500010522	5	36446050	0.50	*TMEM117*	Within	1.45 × 10^−8^	−0.33019
	BovineHD1500023322	15	78715609	0.40	*SBF2*	Within	6.19 × 10^−8^	−0.30795
UD	BTA-75047-no-rs	5	109433376	0.05	*LGALS2*	Within	1.26 × 10^−7^	−0.66283
	BovineHD0600024277	6	86964714	0.21	*GC*	Within	2.92 × 10^−7^	0.321971
	BovineHD0600001885	6	7087395	0.35	*UBE2K*	50 kb	5.16 × 10^−7^	−0.27781
	BovineHD0900001933	9	8043006	0.28	*ADGRB3*	Within	5.98 × 10^−7^	−0.28799
	BovineHD2300001734	23	6986268	0.26	*GCLC*	Within	9.36 × 10^−7^	−0.32781

SNP: single nucleotide polymorphism; CHR: chromosomes; AUA, anterior udder attachment; CSL, central suspensory ligament; PUAH, posterior udder attachment height; PUAW, posterior udder attachment width; UD, udder depth; MAF: minor allele frequency; Effect: each variation regression coefficient.

**Table 4 animals-12-02542-t004:** Details of the KEGG (Kyoto Encyclopedia of Genes and Genomes) pathways analysis enriched from the nearest candidate genes and genes within 200 kb of the significant SNPs.

KEGG ID	Description	Count	Gene Name
bta04978	Mineral absorption	3	*STEAP1, STEAP2, HEPHL1*
bta00480	Glutathione metabolism	3	*MGST1, MGST2, GCLC*
bta00340	Histidine metabolism	2	*HAL, AMDHD1*
bta04935	Growth hormone synthesis, secretion and action	3	*CREB5, SST, CACNA1D*

## Data Availability

In this study, the data presented are available on request from the corresponding author.

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
