# Peer review of "Genome-Wide Association Study for Udder Conformation Traits in Chinese Holstein Cattle"

_animals, 2022, doi:10.3390/ani12192542_

Round 1

Reviewer 1 Report

A brief summary:

The authors conducted GWAS and a pathway analysis to identify genes and pathways associated with udder structure related conformation traits in Chinese Holstein cattle. They also estimated genetic parameters (heritability and genetic correlations) for udder structure related traits in Chinese Holstein cattle.  These genes and pathways could provide targets for selection against mastitis and other bovine mammary infection in Chinese Holstein cattle.

Review:

The major limitation of the study is the readability of the manuscript. English is poor; incoherent sentence and disconnected paragraphs make it hard to follow. It is hard for international audience to understand the text. The authors are strongly recommended to take English language editing services or consult the native English writers to edit the manuscript.

Introduction: The introduction section fails to show context and reason for doing this research. The hypothesis is not well-defined. Why the authors choose to perform this study? It is because the genetic and genomic analysis of udder structure conformation traits are lacking in Chinese Holstein cows, or these traits are proxy to economically important traits like Mastitis which have low heritability and are difficult to improve? A clear hypothesis and implications of the study are expected in the introduction section.

Methods: Why did the author use herd, year, season and parity as random effects in the model? Given these effects have few levels (up to 4), it is best to use them as fixed effects and run a mixed-model equation. Do you think putting all the effects in the model as random effects upwardly biased the heritability estimates? The authors are recommended to re-run the model using herd, year, season and parity as fixed effects and report the genetic parameters. Also. report the standard error of the heritability and correlations.

Results:  The authors are advised to present some follow-up for the positional candidate genes identified in the study. A feasible way of addressing the matter would be to seek NGS data of Holstein cattle from 1000 Bull Genome Project and list putative functional variants (upstream, downstream, missense, stop gain, in frame deletion, splice-variant, etc.) that are polymorphic in Holsteins and that are co-mapping with your GWAS hits. Such a list would be useful for future investigation and could be construed as a minor follow up.

General Comments:

1.    Grammatical errors throughout the manuscript. I have highlighted few. Put the punctuation marks in the appropriate place. The authors are encouraged to take english language editing services.

2.    Do not start a sentence with abbreviated gene name like “XXX”. Instead start with the “XXX”.

Specific Comments:

L23: remove genetic and replace biological with genetic.

L26: re-write the sentence

L35: Provide the SE of heritability estimates

L38: replace connected with significant

L40:  replace Bioinformatics with Pathway

L41: replace pathway analysis with terms

L48: replace has been with is

L49: remove rising

L53: replace effect with affect

L65: remove with solid reasons

L68: Please provide more details regarding how GWAs would accelerate the genetic improvement of quantitative traits in animal breeding.

L70: remove authenticated

L71: replace significant with important

L73-75: complete the sentence

L87-90: I would suggest rephrasing this sentence for increased clarity.

L99: I would suggest writing “no animals were harmed” instead of “no animal felt painful”

L102: remove “was”

L108: remove “were”

L113: How many animals were genotyped? 999 or 1000?

L123: Please take care of the sub-scripts in the equation especially in Parity.

L128: Please include the levels of the effects.

L131: Why did the study consider Herd, Year, Season, and Parity as random effects rather than fixed effects given that these fixed effects have few levels? How would you justify that putting these factor as random effects will over-estimate the heritability components?

L132: replace could be got back with “traced back to”

L144: MAF of SNP < 0.05 (NOT MAF > 0.05) will be removed from the analysis

L170: Why did the study pick up 200 kb position of the candidate genes?

L212-225: I would suggest rephrasing the whole paragraphs for increased clarity.

L231: Please include standard error (SE) for heritability in the table.

L251: remove unable

L271: I would suggest changing “were revealed to visualize” to “revealed”

L273: remove “top of”

L276: replace “were previous” with “passed”

L288: replace noticed with “found”

L299: replace “confirmation: with “conformation”

L305:  Please mention clearly what is on the x-axis and what is on the y-axis. The x-axis displays position of SNP along the chromosomes.

L335: remove “which”

L352: Take care of numbers 353, 354, 356. They are flying.

L379: remove them and replace understand better with “better understand”

L386: replace “are”

L392: remove “was”

L433: replace “has” with “have”

L477: replace “a lot of” with “many”

L480: remove “highest amount”

L501: Different font size. Please be consistent with the font size.

L510: remove “above”

L516: remove comma (,)

L534: replace “serval” with “several

Author Response

Respected sir, I have attached your comments file, kindly, you see it.

Reviewer 2 Report

English must be improved.  Too many sentences are not correct. In many parts of the text, it is not possible to understand the meaning. Therefore, it is not possible to evaluate the scientific merit

Author Response

Response to Reviewer 2 Comments

English must be improved.  Too many sentences are not correct. In many parts of the text, it is not possible to understand the meaning. Therefore, it is not possible to evaluate the scientific merit

Response: Thanks for your comment and we have improved English grammar, English sentences, and disconnected paragraph in all the manuscript.

Reviewer 3 Report

In general, this manuscript is good quality, but contains some faults.

Additional remarks:

Introduction

line 53: suggest „such as” phrase put into brackets!

line56: medium heritable instead of highly heritable!

lines 63 and 72: Udder… and Milk…: please use this word as lower case (milk, etc.)!

line 72: protein? Milk protein? But milk composition includes the milk protein too! Please clarify protein!

Materials and methods

line 103: “The Holstein Cattle included” please delete, after “… China” phrase start the bracket!

line 105: miss a bracket!

line 119: lower case: “Standard” and “Coefficient”!

line 123: in the equation, the “Year” what does it means? No of generation (see lines 132-133) or age of cows? Please clarify it!

line 134: how many seasons were investigated? See line 106! Please clarify it!!

Results

line 199-200: miss the std error from the orders! And I suggest changing the list: mean, std. error, std. deviation, minimum and maximum!

Table 1: please add to the title the number of animals in brackets!

Figure 1: not cited the Figure 1 in the text! Please add explanations from the Figure in the text! Or delete this Figure!

line 250-251: “…the family structure” please refine this sentence!

line 274 and Figure 5b: not cited the Figure 5b in the text! Please add explanations from the Figure in the text! Or delete this Figure!

line 299: please use the “Udder” as lower case in Table 3 title!

line 360: “there is” rather correct to there are!

Discussion

line 370: low and medium instead of average!

line 399: “For the colostrum and serum albumin...” this phrase is not clear! These parameters were not studied! Please re edit this sentence!

line 407: suggest delete this phrase: “… family structures…”!

Suggested correction: exact name of Holstein dairy breed is Holstein-Friesian!

Round 2

Reviewer 1 Report

The authors are thanked for addressing reviewer's comments !

Author Response

Kindly, find the attached reviewer's comments file.

Reviewer 2 Report

The manuscript is still very poor in the English. Even scientific concepts are expressed in a naive language. The manuscript cannot be published in its present form and many improvements are needed.

Many sentences are also disrupted or words are missed or reported twice. The manuscript MUST be revised by English reviewer professional service that is familiar in livestock genomics. In addition, scientific terms, concepts and interpretation of the results MUST be reported in scientifically acceptable ways. At present, many parts are poorly written.

The abstract and the introduction should be revised also in terms of content.

There is confusion in the number of cows considered in the study: 1000, 999, ... it seems that for just one animal, a sample was not taken? 

there is also confusion in the genotyped animals or those included in the the association analysis - how if it was possible to obtain samples for 999 animals figure 2 reports 1000 genotyped cows?

lines 118-122: rephrase in a meaningful way

The way in which hairs were sampled is strange and details of the packing and storing of the specimens are not needed.

All acronyms should be defined: first, the names/words and the the acronyms - not vice versa

As far as I know, the GGP SNP genotyping tool of the Genseek/Neogen is a 150 k. The exact number of targeted SNPs should be reported - that is not 100,000

line 183: Nsnps is not defined

line 185: quantile-quantile plots (QQ-plots)

Statistical analysis: what is pseudoQTN? It is also not clear how stratification was controlled in the model

lines 188-189: the sentence should be rephased - not clear at all: and then why 200 kb and not another window ? from one or from both sides? This window should be calculated based on the level of linkage disequilibrium in the population

Table 1: normal distribution is not followed by some traits - how this was considered in the association analyses?

Minimum and maximum scores are those that are defined in the range - these colums can be eliminated

Table 2 and in many parts: what is reported is the genetic and phenotypic correlation - please revise the text here in many other parts

What is the meaning of the circular manhattan plot? it does not add anything else from the regular ones

Enrichment and the gene network analyses do not make any sense in this study - in the windows where significant SNPs are reported there might be many genes whose role might be irrelevant on the considered traits - these parts must be reduced substantially and related discussion reduced to few lines - this is a pure exercise that does not lead to anything.

Discussion should be substantially reduced - too much speculation is reported 

conclusions are too bold and should be rewritten - stay close to the reported results - and it should not be a summary 

Abstract should be revised accordingly

The title: 

Genome-Wide Association Studies for Udder Conformation Traits in Chinese Holstein Cattle

Author Response

Thanks for your valuable comments. Please find the attached file with responses to your comments. 

Round 3

Reviewer 2 Report

The manuscript still needs many adjustments. 

Line 21: delete "molecular breeding basis"

Abstract: the first sentence does not introduce properly the context.

Introduction was not improved at all. It is still very poor and not appropriate in terms of concepts. It should substantially revise. Below a few examples kn how it should be improved.

Line 49: reduces

Lines 51-54: this sentence does not make any sense. It should be rephrased 

Lines 54-55: udder conformation traits are phenotypes  they are not useful as a selection marker. Please rephrase

Lines 60-61: the sentence is disrupted. Please correct

 Line 63: udder fitness is not the correct term

 Line 65: GWAS do not develop any technique. Please rephrase 

Line 68-69: heritability is mentioned above with an opposite message to what reported here. Please delete or rephrase

Line 70: a gwas cannot improve anything as intended here by thd authors. Please rephrase

Line 71: molecular breeding in humans: NO

Lines 65-71: delete

Line 81: a qtl is not associated, it does explain a fraction of the genetic variability 

Line 84: ...have allowed them... what is referred to "them"?

Lines 94-95: delete

Paragraph 2.2: this is not genotypic data collection 

Line 120: first DNA extraction and then genotyping.

Paragraph 2.3. Change title of this paragraph.

Line 127: genetic correlation

In this paragraph, please use subscrkpts in a correct way for all factors.

Paragraph 2.4 Genotyping not genotypic. The first sentence of this paragraph is repeated above.

Line 143 the GPP tool does not extract DNA!!!

Lines148 150:all filtering criteria should be rewritten. What is written now does not properly indicate the applied filtering strategy.

Lines 152-154: rephrase

Parageaph 2.8.

P value should nog be based on 0.05 but using a multiple testing corrrction: FDR or Bonferroni

Results

Some traits are not normally distributed. Therefore some corrections should be included jn the association analysis

Paragraph 3.3 does not add anything to the result and should be eliminated or just reported in Suppl. Materials with reference to Materials and methods

The legend to figure 2 is wrong. Even if this should be removed, this figure does not report Significance of SNPs.

Line 287: what does it mean?

Line 306: significantly associated

Lines 207-310: this parg of the text is also included in Table 3.

Paragraph 3.6. It should be reduced substantially.  This is just a list of genes.

Paragraphs 3.7 and 3.8 should be completely revised after the adjustment of p value.

Discussion still needs to be shortened.

Author Response

Please, find attachment file
